# Patterns of Egg Consumption Can Help Contribute to Nutrient Recommendations and Are Associated with Diet Quality and Shortfall Nutrient Intakes

**DOI:** 10.3390/nu13114094

**Published:** 2021-11-16

**Authors:** Yanni Papanikolaou, Victor L. Fulgoni

**Affiliations:** 1Nutritional Strategies, 59 Marriott Place, Paris, ON N3L 0A3, Canada; 2Nutrition Impact, 9725 D Drive North, Battle Creek, MI 49014, USA; vic3rd@aol.com

**Keywords:** NHANES, eggs, usual intakes, shortfall nutrients, dietary patterns

## Abstract

Limited data are available on how eggs are consumed in the typical American eating pattern and the contribution to usual intakes, diet quality and in meeting recommendations. The objectives of the present analysis included identifying how eggs are consumed within U.S. dietary patterns and how these patterns are associated with the usual intakes of shortfall nutrients and diet quality (Healthy Eating Index 2015) using data from the combined National Health and Nutrition Examination Survey (NHANES) from 2001–2016. An additional objective included assessing the differences between egg consumers and egg non-consumers in nutrient intakes and nutrient adequacy. Several egg-containing dietary patterns were identified, and two egg patterns were associated with a greater diet quality compared to a no egg pattern (*p* < 0.0001). Most egg patterns identified were similar in diet quality scores when compared to the no egg pattern; however, the two egg patterns had lower diet quality scores. Egg consumption combined with a greater intake of total protein foods, seafood and plant protein, total vegetables, total fruit, whole fruit, whole grains and dairy foods, and a lower intake of refined grains and added sugars contributed to an improved diet quality, supporting that no one food is responsible for a healthy dietary pattern. Egg consumers demonstrated significantly higher intakes of dietary fiber, calcium, magnesium, potassium, total choline, vitamin A, vitamin C, vitamin D and vitamin E when compared to egg non-consumers. A comparison of egg consumers and egg non-consumers found egg consumers had significantly less percentages of the population below the EAR for calcium, iron, magnesium, vitamin A, vitamin C and vitamin E. Similarly, the percentage of the population above the recommendations for potassium and choline were greater for egg consumers vs. egg non-consumers. In egg consumers, 24.4% of the population was above the AI for dietary choline when compared to 4.3% of egg non-consumers (*p* < 0.0001). Findings from the present analysis demonstrate that eggs and egg-containing foods can be an important part of a healthy dietary pattern when balanced accordingly with other nutrient-dense foods.

## 1. Introduction

At present, limited data have been published on how Americans consume eggs within dietary patterns. While previous dietary guidance has been critical of the inclusion of eggs within healthy dietary patterns, the current recommendations from both the preceding 2015–2020 Dietary Guidelines for Americans (2015–2020 DGA) [1] and the current 2020–2025 Dietary Guidelines for Americans (2020–2025 DGA) [2] policy reports support egg consumption within dietary patterns and encourage a variety of nutrient-dense protein foods, whole grains, dairy products and fruits and vegetables. Furthermore, eggs are nutrient-dense foods when prepared with little or no added solid fats, sugars or sodium, thus further aligning with current dietary guidance [1]. Indeed, the 2020–2025 DGA, for the first time in dietary guidance history, has recommend eggs as a critical food for infants and toddlers, in addition to pregnant and lactating women [2], largely due to the choline content provided by eggs [3]. Accumulating evidence has contributed to significant scientific agreement that choline plays an important role in healthy brain development [4,5,6]. One large egg provides approximately 147 mg of total choline and represents one of the richest sources of dietary choline in the diet [3]. Similarly, the 2020–2025 DGA acknowledges that eggs can fit within the recommended eating patterns for children, adolescents and adults due to their high-quality protein, choline and vitamin contributions, with emphasis on shortfall nutrients, including vitamin D [2].

While dietary guidance offers several recommendations for healthy dietary patterns and promotes the inclusion of high-quality protein foods, at present limited evidence is available that assesses the association of different egg-containing food patterns on diet quality and short fall nutrient intakes. Earlier work examining the associations with egg patterns of consumption and cardiovascular disease risk factors, using data from the National Health and Nutrition Examination Survey (NHANES) 2001–2008, identified eight readily consumed egg patterns (including a no egg pattern) in American adults [7]. Findings revealed that cardiovascular disease risk factors can be influenced by other foods groups consumed with eggs (e.g., fast foods), in addition to the baseline health characteristics (e.g., diabetes) and behavioral choices (e.g., smoking), rather than simply focusing on the inclusion of eggs in the diet [7]. Nonetheless, previous work has not examined the associations between egg dietary patterns of consumption, nutrient intakes and diet quality, and whether eggs can be associated with nutrient adequacy or meeting other nutrient requirements. Thus, the objectives of the present study included identifying how eggs are consumed within U.S. dietary patterns and how these patterns are associated with the usual intakes of shortfall nutrients and diet quality using data from the combined NHANES datasets from 2001 through 2016. An additional objective included assessing the differences between egg consumers and egg non-consumers in nutrient intakes and nutrient adequacy. It is hypothesized that eggs and egg-containing meals can be part of healthy dietary patterns and that consumers of certain egg-containing dietary patterns have better nutrient intakes and diet quality.

## 2. Experimental Section

The National Health and Nutrition Examination Survey (NHANES) is a nationally representative, cross-sectional survey of U.S. noninstitutionalized civilian residents. NHANES data are collected by the National Center for Health Statistics of the Centers for Disease Control and Prevention. Full details of the NHANES have been previously and thoroughly documented in the published literature [8,9,10,11,12]. Written informed consent was obtained for all participants or proxies and the survey protocol was approved by the Research Ethics Review Board at the National Center for Health Statistics. Data from eight NHANES datasets (2001–2002; 2003–2004; 2005–2006; 2007–2008; 2009–2010; 2011–2012; 2013–2014; 2015–2016) were combined for the present analysis in individuals ≥2 years of age. Nutrient intake data for the NHANES are from the relevant United States Department of Agriculture (USDA) Food and Nutrient Database for Dietary Studies (FNDDS) [13]. FNDDS are databases that provide the nutrient values for foods and beverages reported in What We Eat in America (WWEIA) [14], the dietary intake component of the NHANES for each data release. The WWEIA Food Categories provide an application to analyze food and beverages as consumed in the American diet. The classification scheme includes over 150 unique categories, and there are 15 main food groups and 46 subcategories of foods.

WWEIA is collected using the Automated Multiple Pass Method (AMPM). The USDA’s AMPM represents a validated dietary data collection instrument that provides an evidence-based, efficient and accurate format for collecting dietary intake data for large-scale national surveys [15]. The AMPM protocol is updated for each 2-year collection of WWEIA to account for the evolving food supply and to address any research needs. The AMPM is a fully computerized recall method that uses a 5-step interview: (1) quick list; (2) forgotten foods; (3) time and occasion; (4) detail cycle; (5) final probe. The AMPM includes an extensive compilation of standardized food-specific questions and possible options [15]. Interviewers use dietary recall status codes in both the Individual Foods and Total Nutrient Intakes files to indicate the validity and reliability of responses (i.e., the quality and completeness of a participant’s responses) [15].

### 2.1. Subjects

In the present analysis, the combined NHANES dataset sample included male and female participants aged ≥2-years-old who were classified as either egg consumers or egg non-consumers. Subjects with reliable and complete 24 h dietary recall data from WWEIA were included in the final analysis (*n* = 65,794). Exclusions included pregnant and lactating females and subjects presenting energy intakes equal to zero. Egg consumers were defined as those consuming eggs (i.e., with the exclusion of mixed dishes) during the 24 h dietary recall. Egg intake was determined using food codes in WWEIA category number 2502 ‘Eggs and omelets’, with exclusions for the FNDDS group number 33 ‘Egg’ substitutes’ and the FNDDS group number 312 ‘other poultry eggs’.

Trained individuals completed the 24 h dietary recalls using the USDA’s AMPM, which includes detailed descriptions of all food and amounts consumed by the subjects. As per WWEIA protocols, all participants were eligible for two 24 h dietary recall interviews. The first dietary recall interview was collected in-person in the Mobile Examination Center (MEC) and the second interview was collected by telephone 3 to 10 days later. Parents or caregivers provided dietary intake information. While 2 days of 24 h dietary recalls were collected in WWEIA, the current analysis used Day 1 data to define egg consumers and egg non-consumers, as this represents the validated in-person data collection.

### 2.2. Methods and Statistical Analysis

Cluster analyses were used to define various dietary patterns that contain eggs and egg products, while intakes of other food groups (total fruits, total vegetables, whole grains, refined grains and protein food groups) further helped to define the dietary patterns. Egg dietary patterns were identified using SAS 9.4 (SAS Institute, Cary, NC, USA) PROC CLUSTER via a 24 h dietary recall in the NHANES. Clusters were developed based on the percentage of calories consumed from eggs and egg-containing foods as the centroid for each cluster. Cluster analyses provided the ability to focus on a particular defined aspect (e.g., calories from eggs and egg-containing foods) and then force maximal differences in clusters for assessment. The patterns identified by the cluster analysis were then identified by the percent of calories of each food group (only groups that contributed 5% or more of calories were used to characterize the clusters).

Total and subcomponents of the Healthy Eating Index 2015 (HEI) scores were used to define diet quality within the dietary patterns. Regression analyses were used to ascertain differences in (1) nutrient adequacy (percentage below the Estimated Average Requirement) and the percentage above the AI of shortfall nutrients as defined previously by dietary guidance [1], including potassium, dietary fiber, choline, magnesium, calcium, iron, and vitamins A, D, E and C, and (2) the total and subcomponent HEI scores. Regression models included age, gender and race/ethnicity as covariates. Usual intakes of shortfall nutrients were determined using the National Cancer Institute’s methodology using both days of dietary recall for calculating the usual intakes [16]. All analyses were adjusted for the complex sample design of the NHANES.

## 3. Results

### 3.1. Usual Energy and Shortfall Nutrient Intakes in Egg Consumers vs. Egg Non-Consumers

Egg consumers consumed approximately 109 additional calories per day when compared to egg non-consumers. However, egg consumers demonstrated significantly higher intakes of dietary fiber, calcium, magnesium, potassium, total choline, vitamin A, vitamin C, vitamin D and vitamin E when compared to egg non-consumers (see Table 1).

### 3.2. Shortfall Nutrient Adequacy and Inadequacies in Egg Consumers vs. Egg Non-Consumers

Comparison of egg consumers and egg non-consumers revealed that egg consumers had significantly less percentages of the population below the EAR for calcium, iron, magnesium, vitamin A, vitamin C and vitamin E (see Table 2). When considering the percentage of the population above the recommendations for potassium and choline, there was a greater percentage of the population above the AI for egg consumers in comparison to egg non-consumers (see Table 2). For dietary choline, 24.4% of egg consumers were above the established AI relative to 4.3% of egg non-consumers (*p* < 0.0001).

### 3.3. Dietary Patterns (Clusters of Egg Consumption) 

Table 3 depicts ten dietary patterns of egg consumption and the corresponding sample sizes. Only patterns of consumption with ≥200 subjects were included in the analysis. Cluster number 0 (*n* = 52,640) represented a no egg consumption dietary pattern. The most popular pattern of consumption (cluster 2 with 10.3% of calories from eggs, *n* = 6981) was characterized by a higher percentage of calories from mixed dishes, snacks/sweets and nonalcoholic beverages than in other patterns. The second most popular pattern (cluster 3 with 8.2% of calories from eggs, *n* = 2390) was higher in protein foods and vegetables than in other patterns, while the third most popular pattern (cluster 4 with 10.4% of calories from eggs, *n* = 1096) was higher in whole grains and fruits than in other patterns.

### 3.4. Egg Dietary Patterns of Consumption and Diet Quality

Table 4 depicts HEI total and subcomponent scores within each egg cluster group. Several egg-containing dietary patterns had total HEI scores that were not significantly different when compared to the no egg dietary pattern. Total HEI scores for clusters 1 and 2 were significantly lower than cluster 0 (no eggs). The cluster 1 dietary pattern had lower intakes from total vegetables, greens and beans, whole grains and dairy, and higher intakes of saturated fat and added sugars when compared to the no eggs pattern. Cluster 2 represented lower intakes of total vegetables, whole grains, seafood and plant protein and added sugars, and greater intakes of saturated fat relative to the no eggs pattern of consumption. In contrast, clusters 3 and 4 had total HEL scores that were significantly greater when compared to cluster 0 (no eggs). Subcomponent HEI scores in cluster 3 were significantly greater for total protein foods, seafood and plant protein, total vegetables, total fruit, whole fruit, whole grains and dairy, and lower intakes of added sugars and refined grains, but also lower for sodium. Individuals in the cluster 4 pattern had 10.4% of total calories sourced from eggs. Cluster 4 individuals also had greater scores from higher consumption of total fruit, whole fruit, whole grains, greens and beans, dairy foods and total protein foods, and higher scores from lower intakes of refined grains, saturated fat and added sugars. This implies that eggs and egg-containing foods can be an important part of a healthy dietary pattern when balanced accordingly with other nutrient-dense food categories.

## 4. Discussion

To our knowledge, this is the first study using the NHANES data to identify egg patterns of consumption in the American population. Our findings identified several egg cluster patterns within the population, with two egg patterns of consumption associated with improved diet quality scores and lower diet quality scores relative to the no egg consumption pattern. All the other egg patterns of consumption that were identified had diet quality scores that were similar to the no egg pattern. The two egg patterns that showed the greatest diet quality scores also had greater intakes of nutrient-dense foods, including greater intakes of total protein foods, seafood and plant protein, total vegetables, total fruit, whole fruit and whole grain, and lower intakes of refined grains and added sugars. These preliminary data imply that eggs can be part of a healthy dietary pattern, particularly an eating pattern that is inclusive of nutrient-rich foods. Furthermore, the current data also support that diet quality changes are not due to eggs, but rather the foods that are consumed with eggs and within the whole of the dietary pattern. The current study also assessed intakes of shortfall nutrients with findings showing that egg consumers have significantly higher intakes of dietary fiber, calcium, magnesium, potassium, total choline, vitamin A, vitamin C, vitamin D and vitamin E when compared to egg non-consumers. Similarly, the data supports that a smaller percentage of egg consumers are below recommendations for calcium, iron, magnesium, vitamin A, vitamin C and vitamin E when compared to egg non-consumers, though some of the changes, while statistically significant, were small. Furthermore, a higher percentage of the population meet the AI for total choline when comparing egg consumers to egg non-consumers.

The present findings align with our previous work and that of other researchers studying eggs, nutrients and health-related outcomes. An NHANES study examining egg consumption and the links to nutrient intakes and growth-related outcomes in an infant population showed egg consumption was associated with higher nutrient intakes when compared to infant egg non-consumers. Specifically, egg consumption was associated with significantly higher intakes of protein, lutein + zeaxanthin, choline, B12, selenium and phosphorus, and lower intakes of added and total sugars per day relative to egg non-consumers [17]. Similarly, our previous NHANES analyses in children and adolescents found that an eating pattern that included eggs was linked with higher amounts of several nutrients, including protein, polyunsaturated and monounsaturated fat, α-linolenic acid, DHA, lutein + zeaxanthin, potassium, phosphorus, choline, riboflavin, selenium, choline and vitamins D, E and A. Egg consumers also consumed less total and added sugars relative to children and adolescents consuming a no egg dietary pattern [18]. 

Previous literature has recognized the dietary significance of choline largely due to its relevance in metabolism and physiology [4,5], with several publications targeting the critical role choline plays in neuronal structures in early life [19,20,21,22]. Choline can be generated endogenously, but the amounts produced do not support physiological requirements [5,23]. Eggs have been identified as a leading dietary source of choline, with a 50 g hard-boiled egg contributing 146.9 mg total choline or 27% of the recommended daily value [3]. The majority of American children consume less than the AI (550 mg for individuals greater than 4 years of age) [5]. It has previously been reported that the average choline intake in children and adolescents was approximately 256 mg per day [5]. A recent modeling analysis by our group has corroborated previous findings that have demonstrated younger populations are not meeting the established recommendations for choline. Concurrently, the same analysis showed that eggs can help to reduce the shortfall gaps in choline intake and improve the likelihood of meeting the recommendations in younger Americans. Overall, modeling the removal of eggs from the diet in younger Americans decreased the intakes of choline, thereby resulting in fewer participants above the AI for choline. In contrast, the addition of eggs to the weekly eating pattern resulted in substantial increases in choline intakes and more individuals above the AI for choline [18]. Likewise, choline plays an important role in adult health, representing an integral role in the structural integrity of cell membranes, methyl metabolism, cholinergic neurotransmission, transmembrane signaling, lipid and cholesterol transport and metabolism [23]. Among U.S. adults ≥19 years old, as well as pregnant women, less than 10% meet the AI for choline [5]. Indeed, U.S. adults consuming eggs were more likely to meet the recommendations for AI relative to egg non-consumers. Furthermore, egg consumers had approximately double the usual intake of choline relative to egg non-consumers. Adults including eggs in their dietary pattern also had slightly elevated diet quality scores [7], thus aligning with the findings of the current analysis.

The current analyses have limitations inherent in observational research and have been reported in numerous previously published, peer-reviewed studies [24,25,26]. The results are dependent on self-reported dietary data for foods, which may involve study participants under- or over-estimating food consumption, thereby leading to inaccuracies in energy and nutrient intakes. Data were also obtained using a 24 h dietary recall, which relies on the study caregiver’s memory, and while the validated methods are used to gather the data, the recall information is subject to inaccuracies and bias from memory challenges and other potential measurement errors experienced in epidemiological investigations [27,28]. Our current analysis considered the dietary patterns with and without egg consumption, so other food choices within an individual’s eating pattern may also contribute to the relationships observed with nutrient intakes. A significant benefit of using the NHANES data for the current analyses include access to a large and nationally representative dataset of adults of various age groups in the U.S. with the corresponding food and nutrient intake data. As the present research is observational, and since growth and development are multifactorial, future research designs will need to consider randomized controlled trials.

## 5. Conclusions

Our analysis has identified that several egg patterns of consumption are routine in the American population. The egg patterns demonstrating the higher diet quality scores compared to a no egg food pattern included greater consumption of nutrient-dense foods, including greens and beans, total fruit, whole fruit, whole grains, dairy, total protein foods and seafood and plant protein. The current data support that eggs can be part of a healthy dietary pattern, particularly an eating pattern that is inclusive of nutrient-dense foods, thereby supporting the notion that foods should not be considered in isolation but rather as part of an entire dietary pattern. Nutrient intake analyses further showed that egg consumers have significantly higher intakes of dietary fiber, calcium, magnesium, potassium, total choline, vitamin A, vitamin C, vitamin D and vitamin E when compared to egg non-consumers. The current data assessing usual intakes also suggest that a smaller percentage of egg consumers are below the recommendations for calcium, iron, magnesium, vitamin A, vitamin C and vitamin E when compared to egg non-consumers. Additionally, when including eggs in the diet, a greater percentage of the population meet the AI for total choline relative to egg non-consumers. The present findings are aligned with the previously published data documenting several benefits associated with egg consumption.

## Figures and Tables

**Table 1 nutrients-13-04094-t001:** Mean intakes for energy and shortfall nutrients in egg consumers vs. egg non-consumers.

Energy/Nutrient	Egg Consumer	Day 1 N	Day 1 Mean	SE	*p*	UI N	UI Mean	SE	*p*-Value
Energy (kcal)	NO	45,791	2082	7.65		45,798	2081	6.67	
Energy (kcal)	YES	20,003	2191	11.95	<0.0001	20,004	2190	10.65	<0.0001
Dietary fiber (gm)	NO	45,791	15.6	0.11		45,798	15.6	0.10	
Dietary fiber (gm)	YES	20,003	16.3	0.14	0.0003	20,004	16.3	0.13	0.0001
Calcium (mg)	NO	45,791	953	5.94		45,798	954	5.18	
Calcium (mg)	YES	20,003	986	7.97	0.0009	20,004	986	7.18	0.0004
Iron (mg)	NO	45,791	15.0	0.08		45,798	15.0	0.07	
Iron (mg)	YES	20,003	15.2	0.11	0.1402	20,004	15.2	0.10	0.1423
Magnesium (mg)	NO	45,791	277	1.66		45,798	277	1.50	
Magnesium (mg)	YES	20,003	297	2.08	<0.0001	20,004	297	1.86	<0.0001
Potassium (mg)	NO	45,791	2521	13.77		45,798	2523	11.87	
Potassium (mg)	YES	20,003	2746	16.36	<0.0001	20,004	2747	14.84	<0.0001
Total choline (mg)	NO	33,500	271	1.69		33,506	270	1.36	
Total choline (mg)	YES	15,673	414	3.18	<0.0001	15,674	412	2.77	<0.0001
Vitamin A, RAE (mcg)	NO	45,791	592	7.67		45,798	591	5.87	
Vitamin A, RAE (mcg)	YES	20,003	685	7.16	<0.0001	20,004	686	6.37	<0.0001
Vitamin C (mg)	NO	45,791	81.4	1.03		45,798	81.1	0.92	
Vitamin C (mg)	YES	20,003	90.1	1.15	<0.0001	20,004	89.7	1.12	<0.0001
Vitamin D (µg)	NO	45,791	4.62	0.04		45,798	4.64	0.04	
Vitamin D (µg)	YES	20,003	5.68	0.07	<0.0001	20,004	5.62	0.06	<0.0001
Vitamin E	NO	45,791	7.38	0.06		45,798	7.37	0.05	
Vitamin E	YES	20,003	8.49	0.10	<0.0001	20,004	8.45	0.09	<0.0001

Data are from Day 1 recall; vitamin D includes D2 + D3; vitamin E is vitamin E as alpha-tocopherol; U.S. individuals ≥2 years old; gender combined; NHANES 2001–2016. From regression analyses comparing consumer/non-consumer with age, gender and race/ethnicity as covariates.

**Table 2 nutrients-13-04094-t002:** Percent of U.S. individuals below the EAR and above the AI in egg consumers vs. egg non-consumers.

Energy/Nutrient	Egg Consumer	<EAR %	SE	*p*	<RDA %	SE	*p*	>AI %	SE	*p*-Value
Energy (kcal)	NO									
Energy (kcal)	YES									
Dietary fiber (gm)	NO							4.94	0.25	
Dietary fiber (gm)	YES							5.49	0.41	0.2494
Calcium (mg)	NO	47.06	0.55		66.58	0.52				
Calcium (mg)	YES	43.54	0.73	0.0001	63.47	0.72	0.0004			
Iron (mg)	NO	4.92	0.19		26.87	0.27				
Iron (mg)	YES	3.07	0.13	<0.0001	22.85	0.30	<0.0001			
Magnesium (mg)	NO	52.61	0.58		71.08	0.50				
Magnesium (mg)	YES	48.28	0.80	<0.0001	68.71	0.68	0.0049			
Potassium (mg)	NO							31.17	0.60	
Potassium (mg)	YES							39.17	0.77	<0.0001
Total choline (mg)	NO							4.31	0.19	
Total choline (mg)	YES							24.41	1.17	<0.0001
Vitamin A, RAE (mcg)	NO	45.55	0.74		70.90	0.69				
Vitamin A, RAE (mcg)	YES	30.72	0.92	<0.0001	63.23	0.83	<0.0001			
Vitamin C (mg)	NO	39.77	0.71		50.70	0.69				
Vitamin C (mg)	YES	34.50	0.81	<0.0001	46.06	0.81	<0.0001			
Vitamin D (µg)	NO	94.64	0.25		99.42	0.06				
Vitamin D (µg)	YES	94.54	0.51	0.8641	99.73	0.07	0.0005			
Vitamin E	NO	86.33	0.47		94.49	0.30				
Vitamin E	YES	81.06	0.83	<0.0001	92.32	0.56	0.0006			

Usual intakes determined using two dietary recalls using the National Cancer Institute method (Tooze et al.). SE = standard error; vitamin D includes D2 + D3; vitamin E is vitamin E as alpha-tocopherol; EAR = estimated average requirement; AI = adequate intake; U.S. individuals ≥2 years old; gender combined; NHANES 2001–2016. From *t*-test comparing consumers/non-consumers.

**Table 3 nutrients-13-04094-t003:** Egg dietary patterns of consumption (clusters) and mean percent calories (kcal) within food groups.

Cluster Number	*n*	Eggs	Dairy	Protein Foods	Mixed Dishes	Refined Grains	Whole Grains	Snacks/Sweets	Fruit	Vegetables	Beverages Nonalcohol	Beverages Alcoholic	Water	Fats/Oils
0	52,460	0.00	8.14	13.42	21.57	10.45	3.30	16.31	2.81	5.23	10.27	3.11	0.07	2.99
1	821	8.68	5.61	13.16	13.91	14.39	1.74	13.16	1.98	4.96	7.37	0.84	0.00	4.37
2	6981	10.33	8.45	10.66	19.59	11.86	1.80	15.92	2.46	3.85	10.72	0.77	0.00	2.38
3	2390	8.21	4.61	26.84	7.55	9.16	1.31	11.75	2.11	13.60	7.54	2.93	0.00	3.10
4	1096	10.38	9.35	12.56	9.36	9.71	12.70	9.85	8.16	5.63	6.68	1.14	0.01	2.63
5	538	7.36	5.93	11.80	14.61	10.71	2.38	12.98	2.01	5.27	5.82	2.42	0.01	17.31
6	790	8.03	4.26	11.99	17.26	7.85	1.54	8.29	0.96	3.83	6.69	25.31	0.01	2.60
7	518	8.84	5.68	13.30	12.41	12.30	2.00	13.53	2.06	5.11	7.60	4.80	0.01	2.89

Data from Day 1 recall from NHANES 2001–2016. Proc Cluster of SAS used to define clusters (only those with sample size >200 retained). WWEAI food groups were used to develop clusters based on % calories from each food group with eggs as the centroid of the cluster.

**Table 4 nutrients-13-04094-t004:** Least square mean total and subcomponent Healthy Eating Index (2015) scores by egg cluster in Americans.

Variable	Cluster 0			Cluster 1			Cluster 2			Cluster 3			Cluster 4			Cluster 5			Cluster 6			Cluster 7		
	LSM	SE	*p*	LSM	SE	*p*	LSM	SE	*p*	LSM	SE	*p*	LSM	SE	*p*	LSM	SE	*p*	LSM	SE	*p*	LSM	SE	*p*
HEI-2015 component 1 (total vegetables)	2.85	0.01	<0.0001	2.33	0.08	<0.0001	2.69	0.04	0.0001	3.48	0.05	<0.0001	2.92	0.09	0.4079	3.39	0.10	<0.0001	2.42	0.08	<0.0001	3.43	0.10	<0.0001
HEI-2015 component 2 (greens and beans)	1.29	0.02	<0.0001	1.00	0.10	<0.0001	1.34	0.05	0.2776	1.45	0.08	0.0572	1.57	0.12	0.0302	1.64	0.16	0.0309	1.26	0.09	0.7433	1.86	0.16	0.0003
HEI-2015 component 3 (total fruit)	2.15	0.02	<0.0001	2.00	0.11	0.1599	2.32	0.05	0.0001	2.00	0.06	0.0130	3.37	0.11	<0.0001	1.95	0.13	0.1360	1.44	0.10	<0.0001	1.99	0.14	0.2495
HEI-2015 component 4 (whole fruit)	2.06	0.02	<0.0001	1.81	0.12	0.0432	2.09	0.06	0.5700	1.87	0.07	0.0032	3.35	0.10	<0.0001	1.72	0.15	0.0297	1.04	0.09	<0.0001	2.00	0.14	0.6698
HEI-2015 component 5 (whole grains)	2.36	0.03	<0.0001	1.68	0.14	<0.0001	1.84	0.06	<0.0001	1.44	0.08	<0.0001	6.61	0.16	<0.0001	1.97	0.17	0.0209	1.44	0.13	<0.0001	2.01	0.19	0.0682
HEI-2015 component 6 (dairy)	5.60	0.03	<0.0001	4.90	0.16	<0.0001	5.94	0.07	<0.0001	4.07	0.10	<0.0001	6.29	0.14	<0.0001	4.99	0.19	0.0030	4.27	0.14	<0.0001	5.33	0.22	0.2240
HEI-2015 component 7 (total protein foods)	3.92	0.01	<0.0001	4.37	0.05	<0.0001	4.46	0.02	<0.0001	4.70	0.02	<0.0001	4.42	0.05	<0.0001	4.36	0.05	<0.0001	4.34	0.05	<0.0001	4.46	0.06	<0.0001
HEI-2015 component 8 (seafood and plant protein)	2.09	0.02	<0.0001	1.86	0.13	0.0723	1.98	0.05	0.0199	2.37	0.08	0.0013	2.20	0.11	0.3152	1.70	0.13	0.0021	1.94	0.11	0.2045	2.37	0.16	0.0784
HEI-2015 component 9 (fatty acid ratio)	4.66	0.03	<0.0001	4.51	0.20	0.4435	4.19	0.07	<0.0001	5.86	0.12	<0.0001	5.09	0.17	0.0123	5.78	0.21	<0.0001	4.64	0.15	0.8882	4.97	0.24	0.1836
HEI-2015 component 10 (sodium)	4.56	0.03	<0.0001	4.46	0.15	0.4901	3.72	0.07	<0.0001	3.33	0.11	<0.0001	4.21	0.18	0.0505	2.94	0.18	<0.0001	5.47	0.17	<0.0001	3.00	0.26	<0.0001
HEI-2015 component 11 (refined grain)	5.80	0.03	<0.0001	6.04	0.20	0.2249	5.87	0.08	0.4011	7.90	0.07	<0.0001	7.96	0.13	<0.0001	6.65	0.20	0.0001	7.59	0.13	<0.0001	6.34	0.23	0.0216
HEI-2015 component 12 (saturated fat)	6.09	0.03	<0.0001	5.16	0.18	<0.0001	4.72	0.07	<0.0001	4.89	0.11	<0.0001	6.63	0.15	0.0004	3.50	0.18	<0.0001	6.84	0.15	<0.0001	4.87	0.23	<0.0001
HEI-2015 component 13 (added sugar)	6.07	0.04	<0.0001	4.78	0.16	<0.0001	6.48	0.09	<0.0001	7.57	0.10	<0.0001	7.61	0.13	<0.0001	7.62	0.17	<0.0001	7.80	0.11	<0.0001	7.30	0.19	<0.0001
HEI-2015 Total Score	49.51	0.16	<0.0001	44.89	0.63	<0.0001	47.63	0.29	<0.0001	50.93	0.40	0.0007	62.24	0.62	<0.0001	48.22	0.83	0.1269	50.49	0.52	0.0645	49.94	0.92	0.6329

LSM = least square mean; SE = standard error; data represent U.S. individuals ≥2 years old; gender combined; NHANES 2001–2016; *n* = 61,140; all *p*-values are compared to cluster 0 (no egg intake); covariates included in analysis were age, gender and ethnicity; *p*-values derived from regression analyses comparing consumer/non-consumer with age, gender and race/ethnicity as covariates.

## Data Availability

Publicly available datasets were analyzed in this study. This data can be found here: https://wwwn.cdc.gov/nchs/nhanes (accessed on 16 July 2021).

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
