# Peer review of "Patterns of Egg Consumption Can Help Contribute to Nutrient Recommendations and Are Associated with Diet Quality and Shortfall Nutrient Intakes"

_nutrients, 2021, doi:10.3390/nu13114094_

Round 1

Reviewer 1 Report

In this manuscript entitled " Patterns of Egg Consumption Can Help Contribute to Nutrient Recommendations and are Associated with Diet Quality and Shortfall Nutrient Intakes ", the authors evaluated how eggs are consumed within US dietary patterns and how these patterns are associated with usual intakes of shortfall nutrients and diet quality using data from combined National Health and Nutrition Examination Survey (NHANES) 2001-2016. The authors eggs and egg-containing foods can be an important part of a healthy dietary pattern when balanced accordingly with other nutrient-dense foods. As stated in this Introduction, the results of this research are very important. In addition, choline, which is found in eggs, is an important nutrient associated with various health benefits and is considered a quasi-essential nutrient. This point is also mentioned in this MS and I judged to be very well organized. On the other hand, the limitations of this study are also discussed. I have a few minor comments, explained below. I hope that my comments are very useful for the improvement of this research.

Comments

(1) Abstract: It is more helpful to the reader if the abstract also describes what the quality of the diet is measured by.

(2) L35-36: It describe “At present, limited data have been published on how Americans consume eggs within dietary patterns.”, but if related content is published, it is better to explain in the introduction.

(3) Table 4: Cluster 8-10 has no HEI score shown. Is the reason for this due to the small number of N? Please explain the reason in the appropriate section.

Author Response

Dear Reviewer,

Thank you for taking the time to review our paper and provide feedback. We have provided answers and comments to your questions and suggestions below. Please let us know if you have any further questions or suggestions. The authors’ answers are in bold font.

Sincerely,

Yanni Papanikolaou and Victor L. Fulgoni

Reviewer’s Question/Comment: Abstract: It is more helpful to the reader if the abstract also describes what the quality of the diet is measured by.

Authors’ Response: Thank you for your feedback. Diet quality was assessed using the USDA’s Healthy Eating Index-2015 scale, which has been previously validated and extensively documented in the literature. Being that there is a word limit with the abstract we have added the term “Healthy Eating Index-2015” in the abstract in brackets to support ‘diet quality’, but have reserved a more detailed description of the Healthy Eating Index-2015 within the manuscript. Please reach back with any further questions/comments.

Reviewer’s Question/Comment: L35-36: It describes “At present, limited data have been published on how Americans consume eggs within dietary patterns…”, but if related content is published, it is better to explain in the introduction.

Authors’ Response: We agree with your comment and this has been included in the 2nd paragraph of the introduction section. The reference to the previously published dietary pattern work reads as follows:

 “Earlier work examining associations with egg patterns of consumption and cardiovascular disease risk factors, using data from the National Health and Nutrition Examination Survey (NHANES) 2001-2008, identified eight readily consumed egg patterns (including a no egg pattern) in American adults [7]. Findings revealed that cardiovascular disease risk factors can be influenced by other foods groups consumed alongside with eggs (i.e., fast foods), in addition to baseline health characteristics (i.e., diabetes) and behavioral choices (i.e., smoking), rather than simply focusing on the inclusion of eggs in the diet [7]. Nonetheless, previous work has not examined associations between egg dietary patterns of consumption, nutrient intakes and diet quality, and whether eggs can be associated with nutrient adequacy or meeting other nutrient requirements.”

Reviewer’s Question/Comment: Table 4: Cluster 8-10 has no HEI score shown. Is the reason for this due to the small number of N? Please explain the reason in the appropriate section.

Authors’ Response: Thank you for highlighting this error. Table 4 is correct and does not include clusters 8-10 due to the small sample size. It is table 3 that is incorrect and has been revised accordingly. Thank you for spending the time and sharing detailed comments to us!

Reviewer 2 Report

The purpose of this research was to identify dietary patterns associated with egg consumption and how these patteres are associated with nutrient intakes and diet quality. The authors did this using NHANES datasets (2001-2016). In general, the paper is well-written and organized. The methods are appropriate and clearly described. The Discussion points to the finding that some egg patterns of consumption are associated with improved diet quality scores, but fairly state that there were patterns that were associated with lower diet quality, suggesting that diet quality may not be due to eggs, but the choice of foods consumed with eggs. This could be an important message. The authors have the expertise to conduct such work. The premise, e.g. that eggs are consumed as part of a dietary pattern is good and this is the direction that nutrition research has been going. The only major comment I have is that there is not a description of the two populations (egg consumers and non egg consumers) and if there is any characteristic differences between these two that may need to be considered. 

Author Response

Dear Reviewer,

Thank you for taking the time to review our paper and provide feedback. We have provided answers and comments to your questions and suggestions below. Please let us know if you have any further questions or suggestions. The authors’ answers are in bold font.

Sincerely,

Yanni Papanikolaou and Victor L. Fulgoni

Reviewer’s Question/Comment: The only major comment I have is that there is not a description of the two populations (egg consumers and non egg consumers) and if there is any characteristic differences between these two that may need to be considered.

Authors’ Response: Thank you for your comment and taking the time to review our paper. We did examine differences between the non-egg consuming dietary pattern and the egg-containing dietary patterns and did not observe any differences. We did see that the greatest contribution of calories in non-egg consumers came from snacks/sweets and mixed dishes, but the differences were not unique or large enough to distinguish from the egg dietary patterns. We are aiming to repeat this analysis in different age groups in the near future, and will continue to look for differences between egg-consumers and non-consumers. Please let us know if you have any further comments/questions.